# Benefits of Native Mycorrhizal Amendments to Perennial Agroecosystems Increases with Field Inoculation Density

**Liz Koziol [1,2,\*], Timothy E. Crews [2] and James D. Bever [1]**

[1] Kansas Biological Station and Ecology and Evolutionary Biology, University of Kansas, Lawrence, KS 66047, USA

[2] The Land Institute, Salina, KS 67041, USA

\* Correspondence: lizkoziol@ku.edu

**Abstract:** Perennial polyculture cropping systems are a novel agroecological approach used to mirror some of the ecological benefits provided by native perennial ecosystems including increased carbon and nitrogen storage, more stable soils, and reduced anthropogenic input. Plants selected for perennial agroecosystems are often closely related to native perennials known to be highly dependent on microbiome biota, such as arbuscular mycorrhizal (AM) fungi. However, most plantings take place in highly disturbed soils where tillage and chemical use may have rendered the AM fungal communities less abundant and ineffective. Studies of mycorrhizal amendments include inoculation densities of 2–10,000 kg of inocula per hectare. These studies report variable results that may depend on inocula volume, composition, or nativeness. Here, we test the response of 19 crop plant species to a native mycorrhizal fungal community in a greenhouse and field experiment. In our field experiment, we chose eight different densities of AM fungal amendment, ranging from 0 to 8192 kg/hectare, representing conventional agricultural practices (no AM fungi addition), commercial product density recommendations, and higher densities more typical of past scientific investigation. We found that plant species that benefited from native mycorrhizal inocula in the greenhouse also benefited from inoculation in the field polyculture planting. However, the densities of mycorrhizal inocula suggested on commercial mycorrhizal products were ineffective, and higher concentrations were required to detect significant benefit plant growth and survival. These data suggest that higher concentrations of mycorrhizal amendment or perhaps alternative distribution methods may be required to utilize native mycorrhizal amendment in agroecology systems.

**Keywords:** arbuscular mycorrhizal fungi; inoculation density; perennial agriculture; plant microbiome

## 1. Introduction

In an effort to regain some of the valuable ecosystem services provided by natural ecosystems, some land managers have begun taking an ecological approach to agricultural practices. One such method is the incorporation of perennial plants and polyculture plantings into cereal and oil seed agroecosystems. The benefit of these perennial polycultures is that they mirror some of the ecological benefits once provided by native perennial ecosystems, such as a prairie grassland. Prairies are diverse assemblages of grasses, legumes, composites, and other major plant groups that remain productive each year with little to no inputs. Prairies created the highly fertile Mollisol soils of the Midwest that have high carbon sequestration [1], soil aggregate stability [2,3], and provide essential habitats for migratory birds, insects, and microbial populations [4]. Intercropped systems have been shown to

have better weed control [5] and increased yields and nitrogen uptake [6], perhaps resulting from plant complementarity [7]. In mimicking some of characteristics of prairies, such as improved plant diversity and minimal soil disturbances, taking a perennial polyculture approach to agriculture will also likely improve soil and ecosystem health and function and lead to more sustainable cropping practices.

Many plant species that are candidates for perennial agroecosystems are derived from or closely related to native grassland species. Many of these prairie grassland species are known to be highly dependent on arbuscular mycorrhizal (AM) fungi [8]. Therefore, it is likely that many perennial crops benefit from mycorrhizal fungi as well. AM fungi are known to improve plant growth by acquiring soil nutrients which are difficult for plants to acquire, such as inorganic phosphorus. AM fungi can also provide non-nutritional benefits to their plant host through alleviation of environmental stressors such as drought [9,10] as well as providing resistance to pathogens [11] and herbivory [12]. AM fungi contribute to other valuable ecosystem services, such as improving soil health and mitigating rising $CO_2$ levels because AM fungi act as carbon sinks [13] and can decrease erosion by producing soil-binding proteins that increase soil aggregate stability [14]. AM fungal abundance is also tightly correlated with nitrogen and carbon sequestration [15]. Due to their many ecosystem contributions, AM fungal communities may be key components utilized in sustainable cropping systems agricultural practices that rely on organic, biologic, or low-input agricultural practices.

Although AM fungi are commonly present in soils, AM fungi in agricultural environments may be limited. The site histories of many agricultural systems include land manipulations known to disrupt fungal communities. For instance, processes such as tilling [16,17], the use of soluble fertilizers and biocides [18], and the planting of crop monocultures [19] can lead to reduced AM fungal abundance, infectivity, and diversity. Reintroducing native AM fungal communities into sites with disturbed soil communities has been shown to benefit grassland plantings by improving plant survival, growth, and fecundity [20,21], and soil aggregate stability [3]. Fungal inoculations seem to be particularly useful when the fungi were isolated from local soils [21,22], likely because fungi are adapted to local nutrient and water conditions [23,24]. However, it remains to be seen whether reintroducing native fungal communities back into disturbed agricultural environments improves the productivity and resilience of perennial polyculture agroecosystems.

While evidence from grassland plantings suggest that AM fungal amendments may also benefit the perennial plants used in polyculture agroecosystems, the recommended application rates and methods provided by commercial producers of AM fungi are highly variable. Inoculation methods commonly include planting pre-inoculated seedlings or broadcasting inocula [25]. Generally, the inoculation rates recommended by commercial producers of inocula range from around 2 kg to 120 kg of inocula per hectare. This volume tends to be significantly lower than what has been reported to be effective application rates from within the scientific literature, which have ranged from an estimated 700–75,000 kg per hectare [20,26–29]. Although lower inoculation rates have been reported to be effective for certain crop species [30], others have found commercial mycorrhizal application to be ineffective [21,22,29]. It remains to be seen whether the ineffectiveness of commercial mycorrhizal products stems from too low of recommended application rates, or from being non-beneficial for other reasons, such as non-native isolates of commercial fungi being maladapted to specific soil or water regimes in which they are applied. Thus, a more thorough assessment of the whether inoculation can be useful in agroecosystems, and at what density, is needed.

In these studies, we test whether mycorrhizal inoculation can improve the establishment and productivity of perennial crop species. First, we conducted a greenhouse study, where we tested the overall mycorrhizal responsiveness of 19 plant species with a locally derived mycorrhizal inoculum. In a paired field study, we designed an inoculation density gradient using the same inocula that covers the range of application rates suggested by commercial producers of mycorrhizal inocula as well as several higher rates corresponding to effective inoculation rates reported in the scientific literature. We followed the establishment and growth response of seven perennial crop candidates selectively bred by The Land Institute (Salina, KS, USA) species that have cereal, oilseed, and legume production potential.

## 2. Materials and Methods

### 2.1. Greenhouse Study

We chose 19 plant species that have cereal, oilseed, and legume production potential to test their response to a native mycorrhizal inocula community (Table S1), including common annual plants and perennial crops. Seeds were obtained from The Land Institute's breeding program or through the U.S. National Plant Germplasm System (NPGS). Plant seeds were germinated in May of 2017 on autoclaved sterilized sand:soil mixtures for a week before being transplanted into a pot containing 2 liters of soil and their mycorrhizal inoculation treatment (living mycorrhiza inocula or sterilized). At this time, initial leaf number was collected for each seedling. Plants were watered twice daily for two minutes via drip irrigation to prevent splashing of soil microbes among treatments. Plants were grown for 8–10 weeks before roots were washed, plant biomass was dried, and weighed for analyses. We had seven replicates of each plant with each inoculation treatment.

A mycorrhizal inoculum was created as described in a previous study [31]. Briefly, single-species fungal cultures were created based on spore morphology from spores isolated from an unploughed, native prairie in Lawrence, Kansas. Cultures were grown for one year in a sterilized sand:soil mixture (10.15 P ppm via Melich extraction, 7.375 NO3-N ppm, 22.2 NH3-N ppm via KCl extractions) in 7-liter pots with *Amorpha canescens*, *Liatris spicata*, and *Andropogon gerardii* as plant hosts prior to being used as inocula. A fungal community mixture of these cultures was homogenized to inoculate our inoculated plant species (*Scutellospora dipurpurescens*, *Gigaspora gigantea*, *Funneliformis mosseae*, *Funneliformis geosporum*, *Glomus mortonii*, *Rhizophagus diaphanous*, and *Claroideoglomus claroideum*). Inoculated plant species were given 50 cm$^3$ of this living inocula, while non-inoculated plant species were given 50 cm$^3$ of sterilized sand:soil mixture used to culture the inocula.

### 2.2. Field Study

This field inoculation study was initiated in the spring of 2017 at the Malone Family Foundation Land Preservation and The Land Institute Gorrill farm located in Lawrence, Kansas, US (39.001311°, −95.320337°). The site was dominated by *Bromus inermis* (smooth brome) that was planted at least 20 years prior to this experiment (Total N 0.19%, Total C 1.92%, 2.67 P-M ppm via Melich, 9.5 ppm NO3-N, 53.69 ppm NH4-N via KCL extraction). In the fall of 2016, the land was disked and tilled twice in early 2017 prior to initiating field plots. Replicate 2x4 meter plots were created with 3 or more meter aisles. Within each plot, each of the seven tested plant species was given one 2 meter row length. Row location was randomized among blocks. We chose 7 plant species that were also included in the greenhouse experiment (Table S1), including legumes alfalfa (*Medicago sativa*), Lupine (*Lupinus polyphyllus*), and Kura clover (*Trifolium ambiguum*), wheatgrasses Heritage Kernza (*Thinopyrum intermedium* in its first selection cycle), Kernza™ 5 (*Thinopyrum intermedium* in its fifth selection cycle), and perennial *Sorghum* hybrid (S. bicolor/S. halepense hybrid) and an oil seed crop candidate Rosinweed (*Silphium integrifolium*). Rosinweed and perennial *Sorghum* seedlings were germinated two weeks prior to the experiment and planted into the field after inoculation, due to low seed recruitment for these species. Then, 2–4 replicates of these seedlings were planted in each plot, and this number was consistent within a block. For the other species, seeding densities of 20 pounds per acre for cereal crops and 12 pounds per acre for other crops were used. Seeds per 2 meter row length were weighed and seeded by hand. All seeds and seedlings were planted immediately after inoculation. During the growing season, weeds between rows were controlled by hand pulling and hoeing. Care was taken to avoid transferring microbes between plots of different inoculation density, including cleaning hand tools, gloves, etc.

Inoculation density treatments were randomized within a block. We had 8 replicate plots for each inoculation density treatment and 16 replicates of the non-inoculated control plots, 2 per block. Mycorrhizae for the field experiment was grown during the 2016 growing season as described for the greenhouse experiment above and utilized the same species. The concentration of inocula was

approximately 30 spores/cm$^3$ or 25,132 spores/kilogram. The number of fungal propagules, which includes infective hyphae, was much greater, though not all fungal species can germinate via fungal propagules such as hyphae and infected root fragments [32] and therefore these propagules were not quantified. Inoculation densities were chosen to represent those used by conventional agriculture (no-inocula), recommended commercial inocula application rates (low-volume inoculation densities), and the inoculation densities used in scientific literature (high-volume inoculation densities) (Table 1). Inocula for each plot was broadcast by hand and then tilled in to the top four inches of the plot with a rototiller. No effort was made to remove the existing mycorrhizal community in the field.

**Table 1.** Eight different inoculation densities were chosen for this study. Densities 1–4 represent those recommended by commercial producers of mycorrhizal amendment products. Densities 5–7 represent a few of the inocula application rates used in the scientific literature.

| Experimental Density | kg/ha | Products/Scientific Studies Using Similar Densities |
|---|---|---|
| 0 | 0 | |
| 1 | 2 | MycoApply (Mycorrhizal Applications) Endo ~2 kg/ha |
| 2 | 8 | Sustainable Agricultural Technologies, INC ~11 kg/ha |
| 3 | 32 | Root Naturally Granular EndoMycorrhize ~24 kg/ha |
| 4 | 128 | MycoBloom Mycorrhizae~168 kg/ha |
| 5 | 512 | Emam 2016 (772 kg/ha whole soil) |
| 6 | 2048 | Koziol and Bever 2017 (1790 kg/ha mycorrhizae) |
| 7 | 8192 | Bever et al. 2003 (10,000+ kg/ha whole soil) |

*2.3. Statistical Analyses*

For the greenhouse experiment, we analyzed the log (1+ total dry weight biomass (g)) of each plant species using Proc GLM in SAS [33]. We used the initial plant leaf number, block, inoculation treatment, plant species, and inoculation treatment*plant species interactions as predictors. Mycorrhizal response was calculated using a ratio of the log transformed weights given by the least squares means as output from our model:

$$\text{Mycorrhizal response} = \frac{\text{Weight with AM fungi inoculation}}{\text{Weight without AM fungi inoculation}}. \tag{1}$$

To calculate the percent change in growth due to mycorrhizal inoculation (Figure 1), we calculated the ((mycorrhizal response × 100) − 100).

For the field experiment, we analyzed the growth and total number of surviving plants of each species during the establishment year. Growth data were transformed as log (1+ plant size). Plant size metric collected varied by species due to their different growth strategies and uses (i.e., fodder vs. seed value). We used total aboveground biomass (g) for alfalfa, total seed weight for perennial *Sorghum*, total tiller or leaf number for Heritage Kernza, Kernza™ 5, and Kura clover, and total leaf number * rosette size for Rosinweed. We analyzed these data using Proc GLM in SAS [33] using block and inoculation density as predictors. To test how plants responded to the different *a priori* categories of inoculation density (none, those reflecting commercial application suggestions, and those used in scientific literature), we broke plant response into 7 contrasts comparing the difference between and among these three categories. We also designed a linear contrast to test if plant response increased with rank of inoculation density. After determining the slope of the response line to rank of inoculation density for both plant growth and plant survival, we tested that slope with that plant's mycorrhizal responsiveness as determined in the greenhouse study as predictor in a linear regression in R [34].

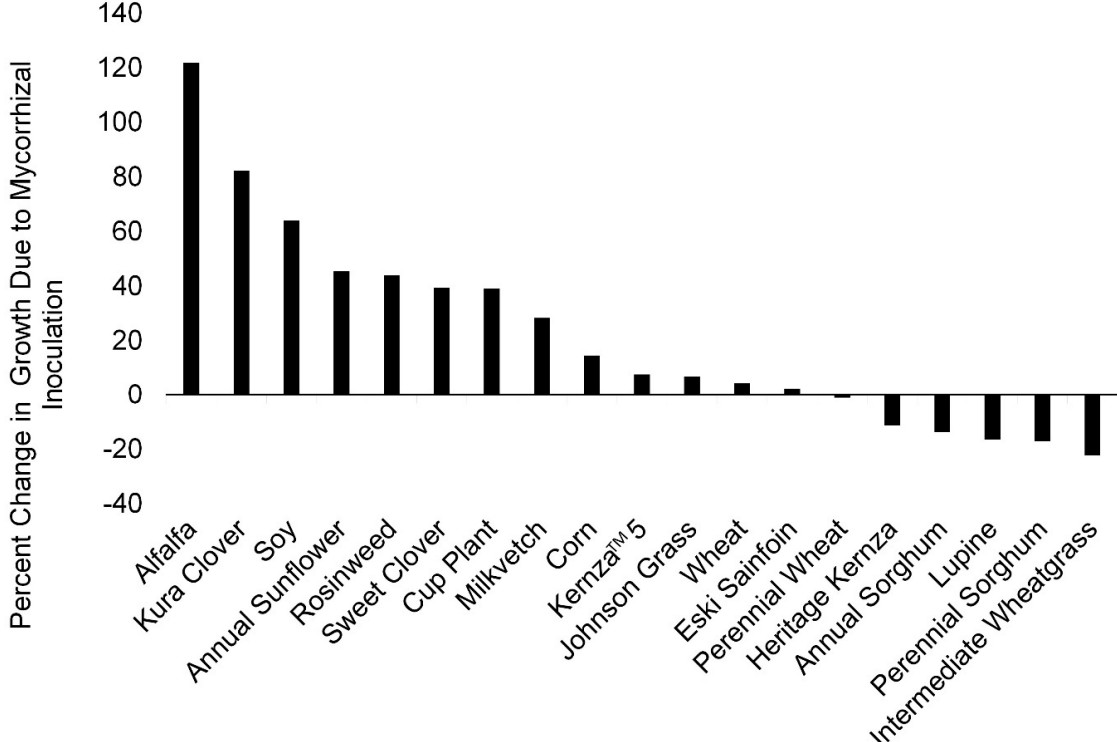

**Figure 1.** The mycorrhizal responsiveness of each of the 19 plant species in the greenhouse study. The growth of many legumes, composites, and some grasses was improved by as much at 120% with inoculation, while some grass species, including many C3 grasses, were unaffected or even inhibited by as much at 17% with mycorrhizal inoculation. Mycorrhizal responsive is a ratio of plant mass when inoculated with arbuscular mycorrhizal (AM) fungi/grown in sterile soil as calculated from the least squared means from our generalized linear model. Bars represent ((mycorrhizal response × 100) − 100).

## 3. Results

### 3.1. Greenhouse Study

Inoculation treatment, plant species, and their interaction were all important predictors of plant growth. We found that across all 19 species, plants grew 14% larger with inoculation ($F_{1,246} = 22.95$, $p < 0.0001$), but varied significantly between plant species (Figure 1, $F_{1,246} = 6.87$, $p < 0.0001$). Alfalfa grew more than 120% larger with inoculation, while other species grew smaller with inoculation, such as a 13% decrease in inoculated plant size for annual *Sorghum* (*Sorghum bicolor*). The top eight mycorrhizal responsive plants out of the 19 tested were composites or legume plant species (Table S1, Figure 1). Four out of the five plants that showed a decrease in growth with inoculation were grass species, while the other was the cluster-rooted plant Lupine (Table S1, Figure 1).

### 3.2. Field Experiment

Plant survival was affected by inoculation rate for some, but not all plant species, and with some, but not all inoculation densities (Figure 2). Across all plant species, plant survival amongst the inoculation rates recommended by commercial producers of mycorrhizal fungi were not found to be significantly different from the controls, suggesting that these densities of AM inocula are not great enough to produce an effect on plant survival using the tested methods (Table 2, commercial vs. non-inoculated contrasts). Several of the plants we found to be generally unaffected by mycorrhizae in the greenhouse study were also unaffected in the field. The survival of Lupine, Kernza™ 5, and Heritage Kernza were not significantly affected by inoculation as evident by an overall non-significant inoculation effect (Table 3), as well as no significant contrasts comparing the different densities of

inocula—except for the survival number of Heritage Kernza seedlings being slightly higher with the highest inoculation densities relative to the controls (Table 3, Figure 2, $F_{1,57}$ = 4.4, $p$ = 0.04). The number of surviving seedlings of Kura clover and alfalfa showed survival improvements in the highest inoculation densities relative to the controls (Table 3, $F_{1,57}$ = 4.34, $p$ = 0.04 and $F_{1,57}$ = 5.37, $p$ = 0.02, respectively) where Kura clover had an average of 1 and alfalfa had an average of three more plants recruited per two-meter row with the highest inoculation rates relative to the controls. The survival of transplanted Rosinweed and perennial *Sorghum* seedlings was generally very high, and few responses to inoculation were found for these plant species (Table 3).

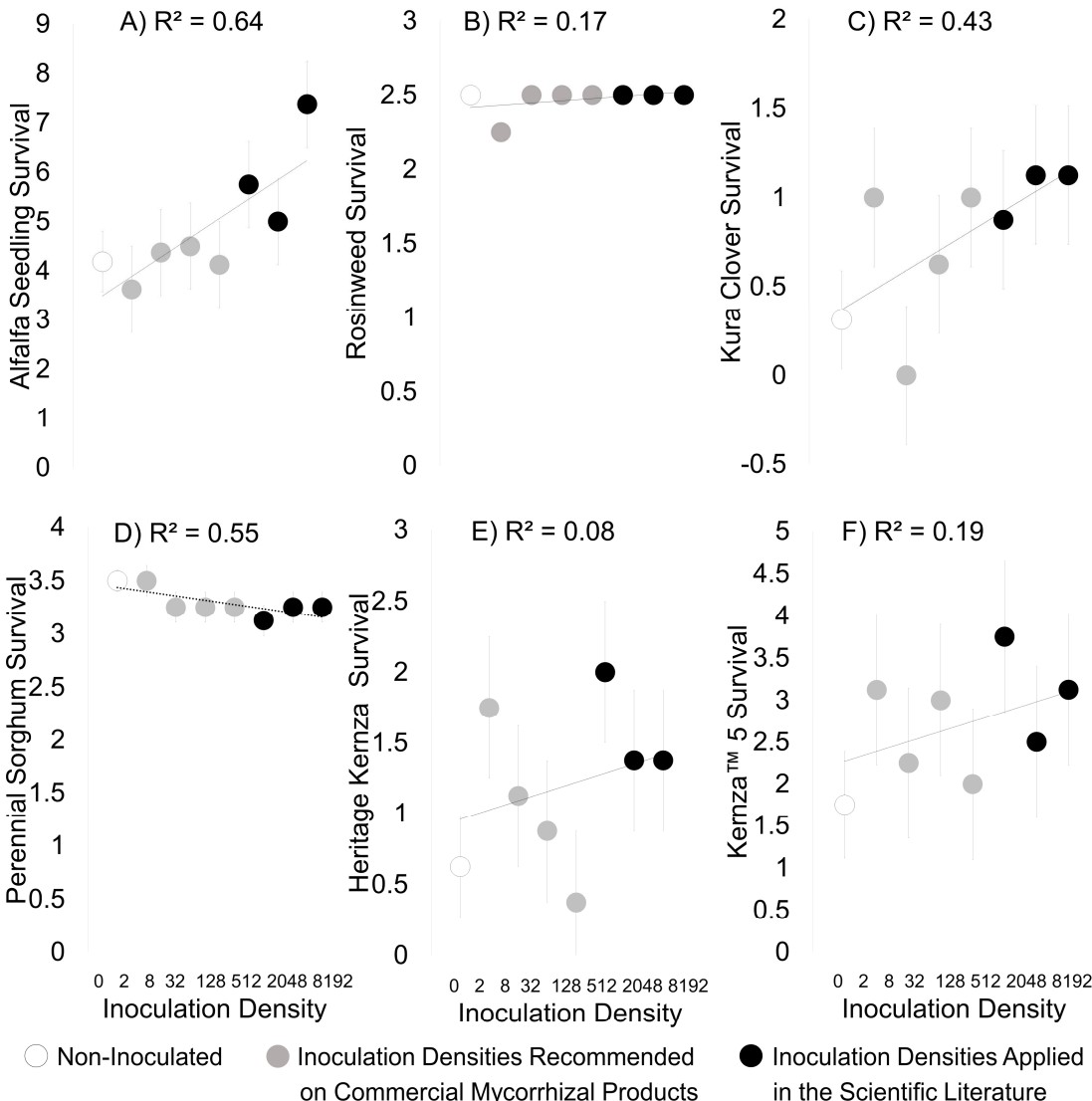

**Figure 2.** Plant survival and establishment in the field when plants were grown with no mycorrhizal inoculation (white, open circles ○), with densities of mycorrhizal amendment recommended by commercial producers if inoculum (grey circles ●), and with mycorrhizal amendment densities that have been shown to be effective in the scientific literature (black circles ●). The survival of some plants was positively (**A**,**C**) or negatively (**B**,**D**) effected by mycorrhizal density, while others were unaffected by mycorrhizal application (**E**,**F**). Dots represent the average plant number when plants were grown with each of the eight inoculation densities, error bars are standard error, and lines are best fit lines for plants that had a linear response relationship with inoculation density.

**Table 2.** Plant survival in the field (number of surviving seedling individuals per plot of each species) was strongly dependent on plot block and for some plant species, also on inoculation density. Because some inoculation densities were predicted to be too low to result in effects on plant growth, we used contrasts within the inoculation term to test whether each plant differed in growth response to different densities of mycorrhizal inoculation, including across and within the categories of having no inocula amendment, adding the low densities recommended by commercial producers of inocula, or adding the high density of mycorrhizal inocula often reported in the scientific literature. A linear contrast was also designed to test if plant response to inoculation was increasing predictably with inoculation density.

| | | Kura Clover | | Alfalfa | | Lupine | | Kernza™ 5 | | Heritage Kernza | | Perennial *Sorghum* | | Rosinweed | |
|---|---|---|---|---|---|---|---|---|---|---|---|---|---|---|---|
| | DF | F-value | *p* | F-value | *p* | F-value | *p* | F-value | *p* | F-value | *p* | F-value | *p* | F-value | *p* |
| **Block** | 7 | 3.68 | 0.002 | 18.01 | <0.0001 | 1.02 | 0.43 | 10.35 | <0.0001 | 6.15 | <0.0001 | 7.79 | <0.0001 | 110.95 | <0.0001 |
| **Inoculation Rate** | 7 | 1.29 | 0.27 | 1.94 | 0.08 | 1.16 | 0.34 | 0.7 | 0.67 | 1.38 | 0.23 | 1.11 | 0.37 | 2.71 | 0.02 |
| Linear Contrast of the Slope of Response to Inocula Densities | 1 | 4 | 0.05 | 9.85 | 0.00 | 0.57 | 0.45 | 0.86 | 0.36 | 0.85 | 0.36 | 3.86 | 0.05 | 3.72 | 0.06 |
| Inoculated vs. Non-Inoculated | 1 | 2.67 | 0.11 | 1.22 | 0.27 | 0.29 | 0.59 | 2.21 | 0.14 | 2.59 | 0.11 | 4.07 | 0.05 | 0.68 | 0.41 |
| Scientific Densities vs. Others | 1 | 4.23 | 0.04 | 5.37 | 0.02 | 1.22 | 0.27 | 0.26 | 0.61 | 0.3 | 0.58 | 2.53 | 0.12 | 2.85 | 0.10 |
| Commercial vs. Scientific Densities | 1 | 2.48 | 0.12 | 5.7 | 0.02 | 1.53 | 0.22 | 0.93 | 0.34 | 3.79 | 0.06 | 0.13 | 0.72 | 0 | 1.00 |
| Commercial vs. Non-Inoculated | 1 | 1.04 | 0.31 | 0 | 0.97 | 0 | 1.00 | 1.18 | 0.28 | 0.89 | 0.35 | 2.28 | 0.14 | 1.78 | 0.19 |
| Scientific Densities vs. Non-Inoculated | 1 | 4.23 | 0.04 | 5.37 | 0.02 | 1.22 | 0.27 | 2.81 | 0.10 | 4.44 | 0.04 | 4.96 | 0.03 | 0 | 1.00 |
| Differences Among Commercial Rates | 3 | 1.47 | 0.23 | 0.2 | 0.90 | 0 | 1.00 | 0.38 | 0.77 | 1.32 | 0.28 | 0.76 | 0.52 | 5.34 | 0.00 |
| Differences Among Scientific Rates | 2 | 0.14 | 0.87 | 1.92 | 0.16 | 3.05 | 0.06 | 0.48 | 0.62 | 0.52 | 0.59 | 0.25 | 0.78 | 0 | 1.00 |

**Table 3.** Plant growth in the field (either plant biomass or size) was strongly dependent on inoculation rate for some plant species, but not others. Contrasts were used to test whether each plant differed in growth response to different densities of mycorrhizal inoculation, including across and within the categories of having no inocula amendment, adding the low densities recommended by commercial producers of inocula, or adding the high density of mycorrhizal inocula often reported in the scientific literature. A linear contrast was also designed to test if plant response to inoculation was increasing predictable with inoculation density.

| | DF | Kura Clover | | Alfalfa | | Lupine | | Kernza™ 5 | | Heritage Kernza | | Perennial *Sorghum* | | Rosinweed | |
|---|---|---|---|---|---|---|---|---|---|---|---|---|---|---|---|
| | | F-value | *p* | F-value | *p* | F-value | *p* | F-value | *p* | F-value | *p* | F-value | *p* | F-value | *p* |
| **Block** | 7 | 2.95 | 0.01 | 12.24 | <0.0001 | 1.02 | 0.43 | 12.02 | <0.0001 | 6.79 | <0.0001 | 14.74 | <0.0001 | 7.85 | <0.0001 |
| **Inoculation Rate** | 7 | 1.36 | 0.24 | 4.37 | 0.0006 | 1.16 | 0.34 | 0.68 | 0.69 | 1.67 | 0.13 | 0.92 | 0.49 | 1.14 | 0.35 |
| Linear Contrast of the Slope of Response to Inocula Densities | 1 | 5.31 | 0.02 | 18.6 | <0.0001 | 0.57 | 0.45 | 1.23 | 0.27 | 0.44 | 0.51 | 2.89 | 0.09 | 4.99 | 0.03 |
| Inoculated vs. Non-Inoculated | 1 | 2.24 | 0.14 | 0.91 | 0.34 | 0.29 | 0.59 | 2.58 | 0.11 | 4.68 | 0.03 | 1.77 | 0.19 | 0.99 | 0.32 |
| Scientific Densities vs. Others | 1 | 5.07 | 0.03 | 8.04 | 0.006 | 1.22 | 0.27 | 0.24 | 0.62 | 0.04 | 0.84 | 2.11 | 0.15 | 2.03 | 0.16 |
| Commercial vs. Scientific Densities | 1 | 4.38 | 0.04 | 9.0 | 0.004 | 1.53 | 0.22 | 1.01 | 0.32 | 1.54 | 0.22 | 0.26 | 0.61 | 1.89 | 0.17 |
| Commercial vs. Non-Inoculated | 1 | 0.39 | 0.54 | 0.29 | 0.59 | 0.0 | 1.00 | 1.47 | 0.23 | 3.03 | 0.09 | 0.96 | 0.33 | 0.05 | 0.83 |
| Scientific Densities vs. Non-Inoculated | 1 | 5.19 | 0.03 | 7.04 | 0.01 | 1.22 | 0.27 | 3.1 | 0.08 | 4.99 | 0.03 | 2.21 | 0.14 | 3.13 | 0.08 |
| Differences Among Commercial Rates | 3 | 1.0 | 0.40 | 1.58 | 0.20 | 0.0 | 1.00 | 0.4 | 0.76 | 2.12 | 0.11 | 1.14 | 0.34 | 0.69 | 0.56 |
| Differences Among Scientific Rates | 2 | 0.12 | 0.89 | 4.98 | 0.01 | 3.05 | 0.06 | 0.22 | 0.81 | 0.1 | 0.91 | 0.42 | 0.66 | 0.7 | 0.50 |

The growth of specific plants the field was strongly affected by inoculation, while other plants were not responsive to native mycorrhizal amendments (Figure 3). As we observed for plant survival, we found that the inoculation rates recommended by commercial producers of mycorrhizae were not different from the controls nor were they different from each other in terms of their effect on plant growth (Table 3, no significant findings under the contrasts comparing commercial vs. non-inoculated or differences among commercial densities). Lupines were not affected by mycorrhizal inoculation density for any metric measured. The growth of Rosinweed, perennial *Sorghum*, Kernza™ 5, Heritage Kernza, Kura clover, and alfalfa were all improved with the highest densities of inoculation relative to the control (Table 3, significant effects or non-significant trends in the scientific densities vs. non-inoculated contrast, Figure 3). We found that the growth of Kura clover, alfalfa, perennial *Sorghum*, and Rosinweed increased significantly with inoculation density.

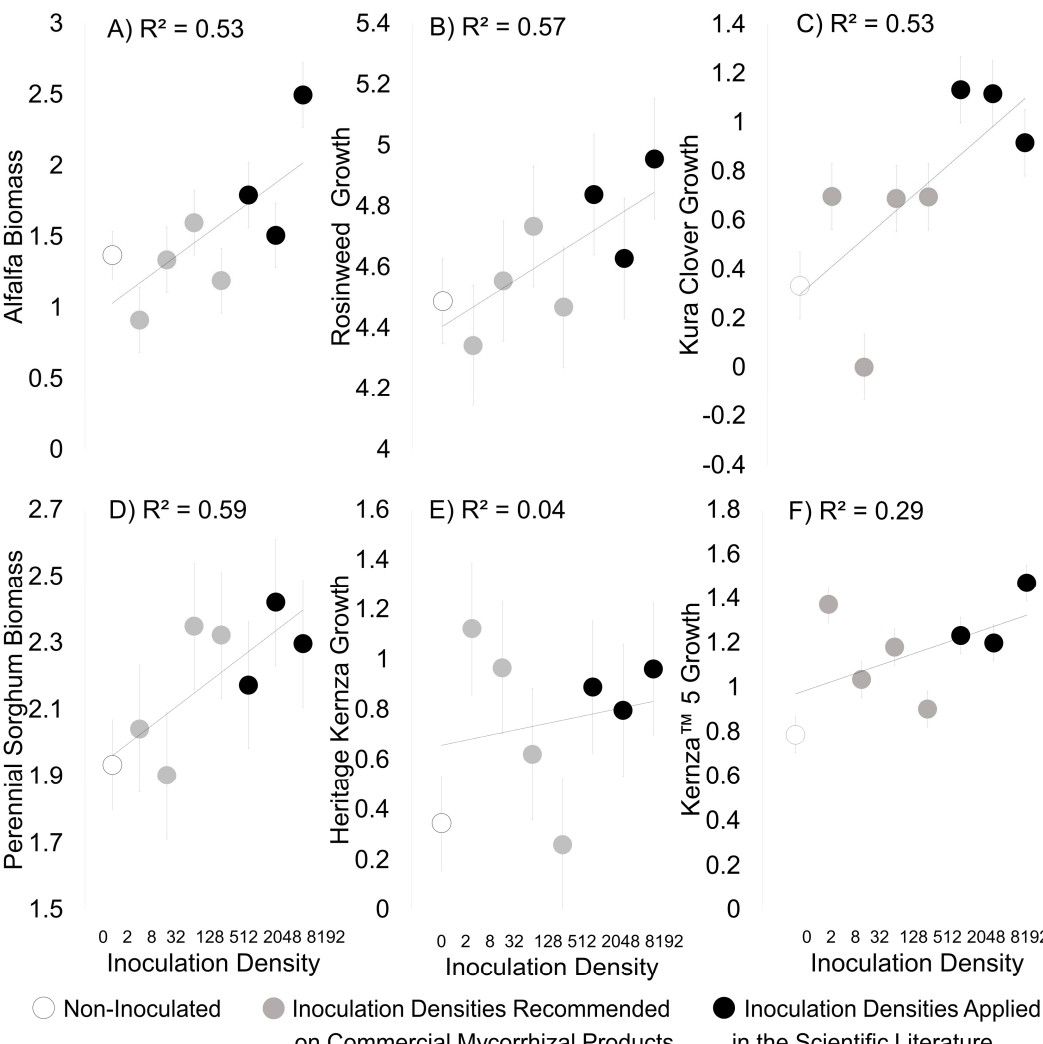

**Figure 3.** Plant growth in the field when plants were grown with no mycorrhizal inoculation (white, open circles ○), with densities of mycorrhizal amendment recommended by commercial producers if inoculum (grey circles ⬤), and with mycorrhizal amendment densities that have been shown to be effective in the scientific literature (black circles ●). The growth of some plants was positively improved (**A**–**D**) or generally unaffected (**E**,**F**) by mycorrhizal density. Dots represent the average plant size when plants were grown with each of the eight inoculation densities, error bars are standard error, and lines are best fit lines for plants that had a linear response relationship with inoculation density.

### 3.3. Field Experiment and Greenhouse Correlations

We found that mycorrhizal responsiveness of individual plant species in the greenhouse was strongly correlated with the response of that plant species to inoculation density in the field (Figure 4). We found the greenhouse mycorrhizal response was 89% correlated with plant growth ($F_{1,5}$ = 19.11 $p$ = 0.007) and was 76% correlated with plant survival ($F_{1,5}$ = 6.74, $p$ = 0.05), indicating that plants that respond well to mycorrhizal inoculation in the greenhouse will likely respond similarly in the field. There was one notable exception. Perennial *Sorghum* had a negative response to mycorrhizal fungi in the greenhouse, growing 17% smaller with inocula. Although high densities of AM fungal inocula resulted in a slight decrease in perennial *sorghum* survival in the field, we found a linear increase in seed production with inoculation density (Table 3, $F_{1,57}$ = 2.89, $p$ = 0.09) where perennial *Sorghum* produced 5–15% more seed in plots with the highest inoculation densities relative to the control.

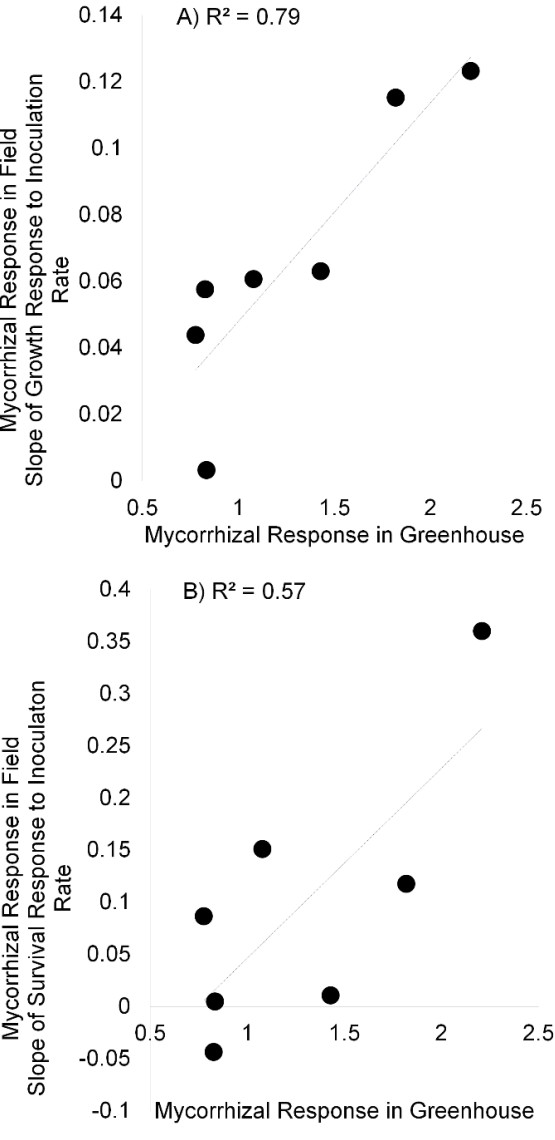

**Figure 4.** Mycorrhizal response (MR, plant mass when inoculated with AM fungi/grown in sterile soil) in the greenhouse was significantly correlated with plant response in the field to the inoculation density gradient (the slope of the line for plant response to the increasing inoculation density) for plant growth (**A**) and plant survival (**B**). Each point represents average MR and slope for a single plant species and lines are best fit lines for each of the two datasets.

## 4. Discussion

These studies demonstrate that manipulations with AM fungi can successfully increase the productivity and survival of several perennial crop candidate species, while having little effect on non-mycorrhizal species, such as the cluster-rooted Lupine plants [35]. In the greenhouse study, we generally found that legumes and composite plant types benefit more strongly from AM fungal inoculation than grass and cereal crop species. Our results are consistent with these previous findings that C3 grasses tend to be less mycorrhizal than other plant functional groups [36], as we found that some C4 grass species and most C3 grasses exhibited slightly negative responses to mycorrhizal inoculation. We also found that plant response to inoculation in the greenhouse was strongly correlated with that plant's response to our increasing mycorrhizal gradient in the field, where non-mycorrhizal species in the greenhouse also did not benefit from mycorrhiza in the field and strongly mycorrhizal plants in the greenhouse were more and more productive with increasing amounts of mycorrhizal inocula in the field polyculture planting. Correlations between plant response to inoculation in the field and greenhouse measures of responsiveness have been observed in previous work in annual and native plant systems [37–39].

*Density of Mycorrhizal Inocula*

As mycorrhizal fungi are abundant in post-agricultural soils in the Midwest, but have markedly different composition from unplowed prairie [40], we interpret the positive response to reintroduction of native mycorrhizal fungi as resulting from a change in AM fungal composition rather than overall mycorrhizal fungal density [23]. Previous field trials have indicated that commercial mycorrhizal fungi are less effective than native, locally adapted mycorrhizal amendments [21,22,29]. Given that commercial AM fungi are likely non-native and that AM fungi can also be adapted to the specific nutrient and water conditions of their soil [23], it is possible that the ineffectiveness of commercial fungi is the result of a mismatch to the soil in which they are being applied. However, commercial fungi application rates are also typically much lower than the density of native mycorrhizal amendments used in scientific studies [26,27,29,31]. Using a common native inocula, we designed our field test to investigate plant response to a mycorrhizal amendment density gradient that represented conventional agricultural practices (no AM fungi addition), commercial density recommendations, and the high inoculation densities that have been found to be successful in scientific studies. Across all plant species that we used in our field study, we were unable to detect a significant inoculation at the rates recommended by commercial producers and that these low densities were generally not different from each other. We found that mycorrhizally responsive plants, as determined by our greenhouse study, were significantly improved with the highest inoculation densities. These results suggest that commercial mycorrhizal amendment application rates may be too low and that more propagules are needed to increase productivity of mycorrhizal plant species. However, it should be noted that our field trial tested inoculation density on short row lengths. Future work should assess the effects of native vs. non-native mycorrhizal amendment (commercial or otherwise) at similar inoculation densities in a perennial cropping system on a larger scale.

Applying mycorrhizal amendments at the densities found to be successful in this study may be cost prohibitive. This study and others that found mycorrhizal amendment to be successful [26,27,29] when inocula is applied via broadcasting and tilling it into the top few inches of soil, a practice that is wasteful of inocula placed in the interrow space. Others have planted inoculated plant seedlings [31], but this method is time consuming. Applications via seed drill have been found to be effective at lower application rates of 25 kg/hectare at the time of planting for corn, but only in conjunction with inorganic fertilizers [30]. More studies are needed on a wider range of crop species. Future work should investigate application methods that more precisely co-locate inoculum with plant roots or seeds, such as drilling via seed drill. Additionally, more work is needed to understand the long-term effects of inocula in perennial agroecosystems, as most of the past work investigating the effects of

mycorrhizal amendment has occurred in annual systems or in grassland restoration systems, where maintaining the productivity or fecundity of specific perennial plant species is not a primary objective.

## 5. Conclusions and Implications to Sustainable Cropping Systems

Incorporating perennial plants and polyculture plantings into agroecosystems has been suggested as a way to mirror the benefits of native perennial ecosystems including increased carbon and nitrogen storage, more stable soils, and reduced anthropogenic input [8]. This body of work demonstrates the importance of ecological approach in sustainable cropping practices by highlighting the importance of the plant microbiome in perennial agricultural systems. Many perennial plant species are strongly dependent on their soil microbes including *Rhizobia* [41], arbuscular mycorrhizal fungi [42], earthworms [43], and a suite of other macro- and microorganisms. Perennial crop species being bred by The Land Institute are newly perennial and generation times are subject to quick turnover. Given that work in grasslands indicate that the beneficial effects of microbial amendments including beneficial mycorrhizal inoculation can increase over time [44], future work should assess if the benefits of amending perennial croplands with symbiotic organisms can outweigh the potential costs of harmful components of the perennial microbiome including pathogen accumulation that can occur in long-living plant species [45]. The plant microbiome may have a particularity important role in polyculture cropping systems. Not only might polycultures prevent the rapid accumulation of soil pathogens, but many choice companion plants are legume species. Many legumes, including several used in this study, are highly dependent on soil mycorrhizae, and may directly or indirectly use them to transfer soil nutrients to companion plants [46]. Taken together, amending new perennial plantings with beneficial microbiome components has the potential to improve plant productivity and establishment. Future work should assess which key soil biotic components to amend with and the optimum planting design for sustainable cropping systems agricultural practices that rely on organic, biologic, or low-input agricultural practices.

**Supplementary Materials:** The following are available online at http://www.mdpi.com/2073-4395/9/7/353/s1, Table S1: Plant Species List, and data and code files: Correlation Tests, Greenhouse, Field Data.

**Author Contributions:** L.K., J.D.B., and T.E.C. conceptualized and reviewed and edited this manuscript. L.K. conduced investigation of the research experiments, visualized data, and wrote the first draft of this manuscript. L.K. and J.D.B. conducted, curated, and analyzed data.

**Funding:** This research was funded by the Perennial Agricultural Project sponsored by the Malone Family Foundation and the Land Institute, the National Science Foundation (DEB-1556664, DEB-1738041, OIA 1656006), and the USDA (grant 2016-67011-25166).

**Acknowledgments:** We would like to thank the Bever/Schultz laboratory group for their contributions to this research.

**Conflicts of Interest:** L.K. is the owner of MycoBloom LLC.

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
