# Peer review of "Benefits of Native Mycorrhizal Amendments to Perennial Agroecosystems Increases with Field Inoculation Density"

_agronomy, doi:10.3390/agronomy9070353_

Round 1
Reviewer 1 Report
This paper discusses mycorrhizal inoculum densities and the effect on different species aimed for perennial and polyculture production. This is a needed area of research, as there is not enough field evidence for the importance of mycorrhizae for plant success, nor the practice of how to achieve mycorrhizal associations that are of agricultural significance.
Overall comments:
- Introduction is overall too long. Should be shortened and made more succinct. Additionally, some text in discussion is misplaced and could or should be part of the introduction.
- More context as to why the particular plant species were chosen. Aside from plants being bred at the Land Institute, why were the others chosen? Are they thought to be companion plants? Or?
- Why didn't the study include a density treatment of the commercial rates of the native "local" mycorrhiza? This would shed a lot of light on the issue discussed in lines 304-320.
- A major oversight or point that needs some discussion is that this study only followed the plants for one year. As these are perennial plants, shouldn't the study have spanned more than one year? Although plant growth and survival may be one indication of plant benefit from the mycorrhizal association, plant establishment, survival over winter, and growth the second and even third year may be significant. Some defense as to why this study was only conducted for one year should be given and later discussion as to how the results might be different if the study was conducted for longer. These results and interpretation carry much less significance since the study was for only one year since the plants of concern are perennial.
Line 13-15: Please reword. In particular "plant candidates" seems like an odd word choice
Line 75-77: Please reword, sentence is too long and reads awkwardly
Line 102-105: Some reason as to why these crops were chosen? Are the field crops chosen different than the 19 species chosen for the greenhouse? Just a bit more detail/clarification here would be helpful
Line 110: reword and clarify this sentence ..."candidates from"??
Line 168-169: This equation is either written incorrectly or not displaying correctly in the journal text
Line 189: "others grew as smaller" ?? reword please
Line 250: misplaced comma
Line 253: "was increasing predictable with inoculation density" -- I don't understand this. Please reword/clarify
Line 282-290: This section is too much of a repeat of the introduction (or what should be in the introduction perhaps) and does not belong in the discussion. You should be leading with the most important findings of the work.
Line 292-295: Sorghum is a C4 grass, so why only mention of C3 grasses here? Why weren't more C4 grasses used in this study?
Line 321-323: The short row lengths used is of concern for proper control measures in the field, especially if there was a strong plot block effect detected (as mentioned in line 249). How do you know if the innocula was contained in a plot? If roots and hyphae formed networks reaching past your row plots? Some more attention and defense of your plot size should be included.
Line 351: Reword including add a missing hyphen: long-lived
Line 354-356: Odd to end the entire manuscript talking specifically of legumes. Ok, but could you end with a broader, over-arching statement about the meaning of your work?
Reviewer 2 Report
Review Agronomy: Benefits of native mycorrhizal amendments to 2perennial agroecosystems increases with field inoculation density . Liz Koziol,* Timothy E. Crews , James D. Bever.
The paper reports the results of two experiments (greenhouse and field) with crop plant species inoculated or not with a mixture of native arbuscular mycorrhizal fungi. The paper provides insights about the responsiveness of the plants to mycorrhizae and the most effective inoculum density in terms of plant growth and survival.
The experimental design, methods used and statistical approach are appropriate to the proposed objectives, which are interesting for the scientific community and perfectly match the scope of the journal.
I only have some minor comments or suggestions:
1. I find sentences in lines 75-77 and 78-79 somewhat contradictory. I there evidence that reintroducing native AM in perenncial polyculture agosystems is beneficial or unknow? Please check and rewrite.
2. Objectives in lines 94-104 can be shortened. For instance, text in line 102 about the effect of inoculation densityh of plant response has been stated before in line 98. Or, it is not necessary here to mention the seven plant species tested in the field experiment.
3. I can not find Table S1 in supplementary material. There are correlation test, fiel data and greenhouse data files instead. A table with the 19 species studied, their latin and common name, their features as perennial/anual, their potential production and so on would be wellcome.
4. Data on mycorrhizal colonization of roots would be essential to understand the results and make conclusión. If available, provide. If not, a comment with this weakness should be added in discussion. Inoculum density is important to plant response, but many papers in literatura demonstated that even with low inoculum densities growth responses can be obtained provided high levels of colonization are reached, and viceversa.
5. Concerning plant names, readers would appreciate authors use the same terminology throughout the text. Sometimes it is common name (figure 2, tables 2,3 ), others latin name with genus and species (figure 1), others only genus (text). Please unify.
6. What plant was used to reproduce and multiply the inocula for the experiments?. What type of containers or pots? I guess big containers taking into account the big amount of inocula needed for the field experiment.
7. The seven plant species should be mentioned in the description of field study in lines 137-138.
8. L 152. Did you determine number of fungal propagules of inocula? If not, please rephrase text in line 152 “was much greater”.
9. Could statistical difference among plant species be shown in figure 1?
10. Why Lupino, Silphium and Shorgum survival data are not represented in figure 2? Similar comment for figure 3.
Author Response
Thank you for your review. Please see attached document.
